# Layer-by-Layer Nanoassemblies for Vaccination Purposes

**DOI:** 10.3390/pharmaceutics15051449

**Published:** 2023-05-10

**Authors:** Eduardo Guzmán, Francisco Ortega, Ramón G. Rubio

**Affiliations:** 1Departamento de Química Física, Facultad de Ciencias Químicas, Universidad Complutense de Madrid, Ciudad Universitaria s/n, 28040 Madrid, Spain; fortega@quim.ucm.es (F.O.); rgrubio@quim.ucm.es (R.G.R.); 2Instituto Pluridisciplinar, Universidad Complutense de Madrid, Paseo Juan XIII, 28040 Madrid, Spain

**Keywords:** immune response, multilayers, Layer-by-Layer, nanomaterials, self-assembly, vaccines

## Abstract

In recent years, the availability of effective vaccines has become a public health challenge due to the proliferation of different pandemic outbreaks which are a risk for the world population health. Therefore, the manufacturing of new formulations providing a robust immune response against specific diseases is of paramount importance. This can be partially faced by introducing vaccination systems based on nanostructured materials, and in particular, nanoassemblies obtained by the Layer-by-Layer (LbL) method. This has emerged, in recent years, as a very promising alternative for the design and optimization of effective vaccination platforms. In particular, the versatility and modularity of the LbL method provide very powerful tools for fabricating functional materials, opening new avenues on the design of different biomedical tools, including very specific vaccination platforms. Moreover, the possibility to control the shape, size, and chemical composition of the supramolecular nanoassemblies obtained by the LbL method offers new opportunities for manufacturing materials which can be administered following specific routes and present very specific targeting. Thus, it will be possible to increase the patient convenience and the efficacy of the vaccination programs. This review presents a general overview on the state of the art of the fabrication of vaccination platforms based on LbL materials, trying to highlight some important advantages offered by these systems.

## 1. Introduction

The Layer-by-Layer (LbL) method is a versatile tool for the fabrication of multifunctional nanomaterials, taking advantage of its simplicity for the assembly of supramolecular systems with a broad range of physicochemical properties and applications [1,2]. This has stimulated an extensive research activity trying to exploit the LbL method for the fabrication of functional nanoassemblies which find application in different technological areas [3,4,5]. For instance, biomedical and pharmaceutical industries have extensively used the LbL method for the design and fabrication of a broad range of nanomaterials with applications in therapy and sensing [4,6,7,8]. Moreover, novel theranostic devices have also been assembled by taking advantage of the modularity and versatility of the LbL method [9,10]. In addition, the versatility and modularity of LbL materials are key aspects for its exploitation as a very powerful tool for the manufacturing of a new generation of nanovaccines [11].

The interest in the use of LbL materials in the fabrication of vaccines has been stimulated for the possibility that this methodology offers to fabricate well-sketched vectors in which their characteristics, including size, shape, surface properties, and composition [12], can be modulated almost at will. This provides the basis for protecting immunogenic cargos in a controlled environment, producing a long-lasting stimulatory effect on the immune systems [13,14]. Moreover, these vaccination platforms can contribute to boosting the humoral and cell-mediated immunity. These promising characteristics have contributed to the development of the LbL method for obtaining versatile vaccination platforms, including nasal drops or microneedles [15,16,17]. Moreover, LbL vaccines can be manufactured with a fine control over some important physico-chemical properties that are of paramount importance for the in vivo application of vaccines [12].

This review is intended to present a general overview of the potential interest of LbL materials to the development of vaccination platforms, providing an updated perspective on the potential interest in the use of type of nanoassemblies as a tool for improving the performance of vaccines. It is expected that this aim can only be achieved by a careful analysis of the main aspects governing the assembly of LbL nanomaterials as well as of the properties of the obtained nanoassemblies. For this purpose, this review includes a short introduction to the LbL method before paying attention to its application in the design of vaccines based on assembled nanostructures.

## 2. A Brief Introduction to the LbL Method

The LbL method was introduced by Decher et al. [18,19,20,21] more than 30 years ago for the assembly of alternate layers of oppositely charged molecules, commonly polyelectrolytes bearing oppositely charges or bolaamphiphiles on macroscopic substrate, but this technique was quickly extended to other families of charged compounds, and even to the assembly of chemical compounds without being charged. Therefore, the existence of complementary interactions between the molecules forming the alternate deposited layers is the main requirement for the fabrication of LbL nanoassemblies, and today, it is possible to find LbL materials assembled through hydrogen bonding, charge transfer interactions, molecular recognition coordination interactions, chiral recognition, host–guest interactions, π–π interactions, biospecific interactions, sol–gel reactions, or even covalent bond (“click chemistry” reactions) [22,23]. The only requirement is a driving interaction strong enough to guarantee both chemical and mechanical integrity of the LbL material upon their exposure to stresses. This makes it possible to manufacture LbL nanoassemblies with a broad range of materials, e.g., synthetic oppositely charged polyelectrolytes (both strong and weak polyelectrolytes); synthetic neutral polymers; colloidal particles and nano-objects (graphene and graphene oxide nanoplatelets, carbon nanotubes, dendrimers, clays, microgels, polymeric, ceramic, or metallic particles); biomolecules (proteins and peptides, polysaccharides, nucleic acid, lipids); dyes; viruses; and even, in some cases, molecular species.

Moreover, the LbL method can be exploited for coating any kind of substrate, independently of their size, shape, or chemical nature, which contributes to increase the versatility and impact of this methodology [1]. In fact, even though the LbL method was initially introduced for manufacturing films using as substrates, or templates, traditional flat macroscopic surfaces, including silicon wafers, quartz plates, or glass slides, among others, it was quickly developed to manufacture nanostructured materials in any solvent accessible substrate, which makes it possible to use as substrates colloidal micro- and nanoparticles, liposomes or vesicles, micelles, fluid interfaces (floating multilayers), emulsion droplets, or even cells [24,25,26,27,28,29,30,31,32,33,34]. It should be noted that the substrate chosen as template for the assembly of LbL materials can play different roles depending on their properties. For instance, the substrate can determine the geometry and morphology of the LbL nano-assembly, remaining as a scaffold in the obtained material, or can simply be a template (sacrificial template) that can be removed from the final material at the end of the fabrication process by using chemical or physical procedures which leads to the production of free-standing LbL nanoassemblies [35,36]. It is worth mentioning that the manufacture of LbL materials is only possible by following suitable fabrication protocols. The most common approaches for the fabrication of LbL materials follow a strategy reminiscent from the methodology introduced by Decher et al. [19], which is based in the alternate dipping of the substrates into solutions containing the components to be assembled. However, the application of this strategy is not always trivial, and requires specific adaptions to consider the characteristics of the used substrate. Therefore, the traditional dipping methodology requires slight modifications to make the fabrication of a broad range of novel materials with different morphologies, sizes, or chemical nature possible [37,38,39,40,41,42,43,44,45,46,47,48,49,50,51,52,53,54,55,56,57,58,59,60,61,62]. Table 1 summarizes some of the most common strategies used for the assembly of LbL materials depending on the nature of the substrate.

The LbL method should be considered a low-cost methodology with a high level of modularity, versatility, and simplicity, which has opened important avenues for its use on the fabrication of multi-layered nanostructured materials with tailored structure, well-defined thickness, and composition. Moreover, the LbL approach enables the introduction of different functionalities in the final material, which is very useful to include specific chemical, biological, optical, mechanical, or electrical properties in the manufactured materials [23,63,64,65]. These aspects have contributed to the exploitation of the LbL approach for manufacturing materials with different characteristics, including flat films, nano- and micro-capsules, or even multicapsules formed by several hierarchically organized nanocontainers [23,30,66,67,68,69,70]. Moreover, the LbL method provides the bases for the fabrication of very complex supramolecular nanoassemblies, e.g., particles with onion-like structures, sponges, membranes, or nanotubes [71,72,73]. This has stimulated a rapid growth of the application of LbL materials in different areas of science and technology, e.g., antibacterial coatings, scaffold for tissue engineering, biomedical devices, wound healing dressing, encapsulation platforms, biosensors, free-standing membranes, protective coatings, self-healing and superhydrophobic surfaces, biosensors and biofuel cells, or materials for photonic applications, among others [18,63,73,74,75,76,77,78,79]. In fact, there are several commercial products based on the LbL technology, including contact lens coated by LbL films (Ciba Vision, Duluth, GE, USA) and coatings for chromatography columns (Agilent Technologies, Santa Clara, CA, USA) [79]. Figure 1a summarizes some of the most relevant aspects of the LbL methods, including their merits and characteristics.

To date, there is a deep theoretical and experimental knowledge about different aspects of LbL materials, including the dependence of the adsorbed amount on the number of adsorbed bilayers, and their potential application. However, there are other aspects, e.g., physico-chemical bases, driving the assembly of LbL materials and their properties, that are less understood, even though they are of paramount importance for establishing correlations between the internal structure and molecular properties of LbL nanoassemblies [33,80,81]. These are strongly dependent on the different variables controlling the assembly process of LbL materials, e.g., charge density of the assembled compounds and of the substrates, concentration, polyelectrolyte molecular weights, ionic strength, solvent quality for the building blocks, pH, and temperature [82]. In fact, all of the above parameters play a very important role in the final structure of the films and in controlling the film stratification. This is important because in most cases, LbL nanoassemblies are not truly stratified systems, presenting a high degree of intermingling between adjacent layers. Further details on the stratification of LbL films are discussed in our previous work [1]. Figure 1b presents a summary of different parameters affecting the assembly of LbL films.

**Figure 1 pharmaceutics-15-01449-f001:**
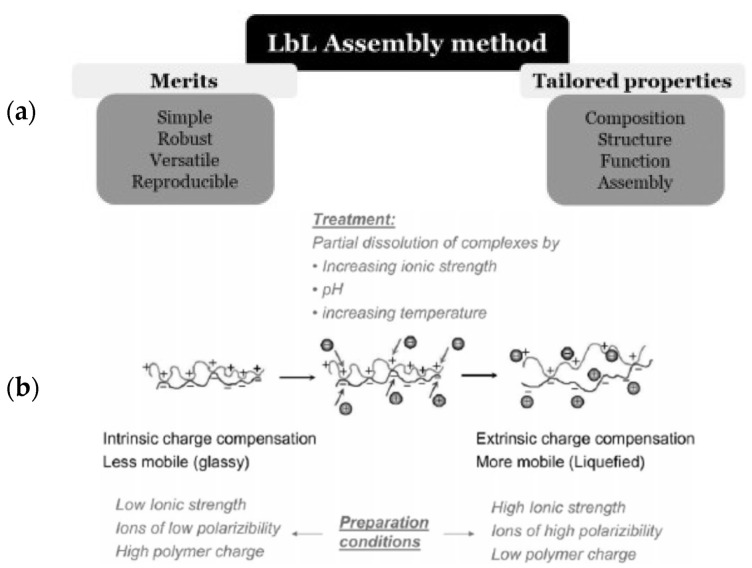
(**a**) Scheme of the main merits and characteristics of the LbL method for the fabrication of structured nanoassemblies. Reprinted from Guzmán et al. [1]. Copyright (2020), with permission from Elsevier. (**b**) Summary of the different parameters affecting the assembly of LbL films. Reproduced from with permission from [82] the PCCP Owner Societies.

It should be noted that the understanding on how the above parameters allow tuning the structure of the multilayered nanoassemblies is of a paramount importance, especially because they may provide the bases for obtaining materials with specific physico-chemical properties and functionalities. This has allowed the fabrication of both passive and active LbL nanomaterials with a broad range of applications [65,74], taking advantage of the ability of LbL for response to external stimuli, e.g., physical (ultrasound, magnetic field, laser pulse, or optical radiation), chemical (the pH, ionic strength or polarity of the environment), or biochemical (receptors or target cells) stimuli [83,84,85,86,87]. Unfortunately, the broadness of the research on LbL films makes it very difficult to present a comprehensive revision here. The reader can find a more complete overview on the fabrication of LbL materials in previous publications [1,7,88].

## 3. LbL Flat Films for Vaccination Purposes

The use of LbL flat films as real vaccination platforms has been limited for long time because this type of supramolecular architecture can only be used as strips for topical application, which in many cases is not a very effective route for vaccination purposes. Therefore, many studies related to the design of vaccines based on LbL flat films are aimed at providing a proof of concept for the fabrication of more complex architectures with real applications. However, in recent years, a promising practical application for LbL flat films as vaccination platforms has emerged in the form of sublingual patches [89]. This type of systems may open new avenues for the exploitation of LbL flat films for vaccination purposes.

Su et al. [90] designed LbL films based on the alternate assembly of layers of a cationic poly(β-amino ester), and ovalbumin as model protein antigen and/or immunostimulatory cytosine−phosphate diester−guanine-rich DNA oligonucleotide adjuvant molecules (CpG). These films can undergo a fast deconstruction upon treatment with a saline solution, resulting in the release of functional ovalbumin. This is a signature of the ability of LbL films for preserving the structure of biomolecules upon encapsulation. The application of the films on transgenic mouse model in which epidermal antigen presenting cells can express green fluorescent protein-tagged major histocompatibility complex molecules demonstrated that the protein antigen released from the films upon application as patches is efficiently acquired by the immune skin cells after several hours of application. Moreover, the loading of antigen and adjuvant molecules together in a single platform enabled a temporal control of the immune response as result of the different release kinetics of each type of molecule.

Monge et al. [89] proposed the design of LbL mucoadhesive patches formed by alternate layers of chitosan and hyaluronic acid for a mucosal administration of vaccines. The fabricated patches were capped with adjuvant particles carrying viral proteins. These were released upon application crossing the most external layers of the sublingual mucosa to reach the epithelium. Moreover, the activity of the encapsulated adjuvants was maintained upon their inclusion in the supramolecular architecture, enabling the activation of the immune response.

## 4. LbL Capsules for Vaccination

Vaccines based on LbL capsules (both core-shell and hollow ones) are commonly used to induce a potent immune response without using adjuvant. Moreover, it is also possible to produce vaccines based on LbL capsules in which the immunogenic cargo is not actually encapsulated but forms an outer coating layer of an empty capsule. The use of this type of system is possible by taking advantage of the ability of immune cells to recognize the structure of these supramolecular nanoassemblies, which mimic some of the morphological features of pathogens, e.g., viruses and bacteria [91]. The versatility of the LbL methods offers a very promising alternative for the fabrication of cell- or virus-like particles which can be exploited for the transport and release of biomolecules with different medical purposes. In fact, in recent years, the study of the use of LbL capsules for the fabrication of vaccines enabling the delivery of antigens and adjuvants to immune cells has undergone a spectacular development [92,93]. This has been possible due to some of the advantages offered by LbL capsules as vaccination platforms, including: (i) the possibility to obtain modular nanoassemblies of different antigens and adjuvants; (ii) the ability for control the loading of components in the supramolecular system; (iii) the efficiency for protecting encapsulated molecules from denaturing conditions, and (iv) the low energy required for manufacturing LbL materials. These characteristics were exploited by Powell et al. [94] for the fabrication of core-shell capsules by the alternate deposition of layers of poly(L-glutamic acid) (PLGA), and poly(L-lysine) (PLL) on solid CaCO_3_ cores, including, after the assembly of a multilayer containing 5 polypeptide bilayers, a capping layer formed by epitopes of the circumsporozoite (CS) protein of *Plasmodium falciparum.* Thus, it was possible to produce synthetic vaccines against malaria containing a tri-epitope CS peptide (T1BT**)* formed by the antibody epitope of the CS repeat region (B) and two T-cell epitopes, the highly conserved T1 epitope and the universal epitope T*. The application of such vaccine particles in mice contributes to the production of parasite-neutralizing antibodies and malaria-specific T-Cell response, including cytotoxic effector T-cells. Therefore, this type of vaccine elicited neutralizing antibodies and effector T-cells specific for the CS epitopes, providing protection to the treated individuals against mosquito challenge with *Plasmodium* sporozoites expressing *Plasmodium falciparum* CS repeats. Following a similar protocol, Powell et al. [91] fabricated LbL nanoparticles loaded with ovalbumin. These particles can be efficiently internalized by dendritic cells, leading to their maturation without secretion of inflammatory cytokines. Moreover, the immunization of mice using the fabricated particles guarantees strong T-cell response, providing protection to immunized mice from challenge with *Listeria monocytogenes.* On the other hand, the administration of the vaccine based in LbL particles elicited an immune response without any side effect of the rest of the components of the vaccination platform.

Jorquera et al. [95] designed a LbL particle as prophylactic treatment against the respiratory syncytial virus (RSV). They assembled LbL structures including respiratory syncytial virus G glycoprotein CX3C motifs which present strong therapeutic effect, contributing to virus clearance and the decrease of the leukocyte trafficking to the lung of infected individuals. The loading of the G glycoprotein CX3C chemokine motifs in LbL particles induces the production of blocking antibodies which inhibit the interaction between the respiratory syncytial virus G protein and the fractalkine receptor (CX3CR1). This contributes to the protection against virus replication and pathogenesis, and hence the vaccination with this type of platforms minimizes the impact of the infection by respiratory syncytial virus disease. Further studies on the effectiveness of LbL particles for the fabrication of vaccines against the RSV were also performed by Jorquera et al. [96]. They found that the vaccination of mice with particles incorporating the RSV G glycoprotein with CX3C motifs inhibits the replication of RSV in the lungs upon infection. Moreover, vaccinated mice present higher levels of RSV G protein-specific IL-4 and interferon gamma (IFN-γ) secreting cells in comparison with control populations, resulting in a potent neutralizing antibody response and reduction of the RSV pathogenesis.

The study of the use of LbL particles for the vaccination against the respiratory syncytial virus was further extended by Powell et al. [97]. They extended the system fabricated by Jorquera et al. [95] by including CD8 epitope of the virus matrix protein (GM2) with or without covalently linked TLR2 agonist (Pam3.GM2). This type of formulation elicited G-specific antibody response and M2-specific CD8^+^ T-cell response upon administration in mice. Moreover, after the exposure to the respiratory syncytial virus, the immunized mice with vaccines without Pam3.GM2 present a Th2-biased immune response in the lungs characterized by high levels of IL-4, IL-5, IL-13, and eotaxin in the bronchoalveolar lavage (BAL) together with a pulmonary influx of eosinophils. On the other hand, immunization with vaccines including Pam3.GM2 leads to non-detectable levels of Th2 cytokines and chemokines, and a low number of eosinophils in the BAL after exposure to the virus. Moreover, the use of vaccines with Pam3.GM2 ensures higher levels of Immunoglobulin G in the BAL after exposure. Therefore, even though both types of formulations are good alternatives for providing protection against the respiratory syncytial virus, the inclusion of TLR2 agonist (Pam3.GM2) provides a more potent antibody response, higher levels of protection, and a clear shift away from Th2/eosinophil response. To increase the immune response associated with the vaccination using LbL particles, De Geest et al. [98] proposed the fabrication of LbL hollow capsules of dextran sulphate and poly-L-arginine decorated with immune-activating CpG-containing oligonucleotides and filled with ovalbumin. This combination ensures a better immunization than capsules containing only encapsulated ovalbumin or a mixture of the later and CpG-containing oligonucleotides.

Sexton et al. [99] designed LbL vaccines by encapsulating a solid core decorated with an antigen protein (ovalbumin) and coated by the alternate deposition of poly(methacrylic acid) modified with thiol groups (PMA_SH_) and poly(vinylpyrrolidone). Thus, it was possible to obtain hydrogel capsules for vaccination purposes by exploiting the covalent crosslinking of thiol group resulting in the formation of disulfide bonds. These hydrogel capsules are internalized by mouse antigen-presenting cells (APCs) in such a way that leads to the presentation of ovalbumin epitopes, leading to the activation of ovalbumin-specific CD4 and CD8 T-cells. The proliferation of these cells is 70-fold and 6-fold higher for CD4 and CD8 T-cells, respectively, than that obtained upon vaccination with an equivalent amount of free ovalbumin. This is a very important signature of the effect of nanoengineered LbL particles as tool for vaccine vehiculation and targeting. Moreover, the work by Sexton et al. [99] also demonstrated that the size of the particles used for vaccination plays a very important role in the control of the immune response. Thus, even though particles with diameter in the range 0.5–1 μm induce the proliferation of T-cells, the higher the particle size, the higher the immunization level. This may be rationalized considering that the smaller the particle diameter, the lower the surface area available for the adsorption of the antigen, and hence the lower the administered dose. Chong et al. [100] substitute the model antigen (ovalbumin) used in the work by Sexton et al. [99] for a KP9 oligopeptide, and demonstrated that the vaccination with LbL hydrogel capsules loaded with the mentioned oligopeptide was fully successful in delivering the cargo in the APCs, activating CD8 T lymphocytes in a non-human primate model of Simian immunodeficiency virus (SIV) infection.

The use of a time-controlled pulsatile release of antigenic material has recently gained importance as a new vaccination strategy. Wang et al. [101] fabricated vaccines based in a model protein antigen ovalbumin, OVA, in a multilayered structure, reaching an antigen release with predetermined intervals. Therefore, it was possible to obtain a release profile mimicking that obtained for multi-dose immunization programs, using a single-injection vaccine. This leads to potent humoral and cellular immune responses, and a long-term persistence of the immunization, reaching stronger immunity levels than those obtained with vaccination programs based on the application of several doses. Figure 2 shows sketches of the fabrication process of single-injection vaccines, and the time-controlled pulsatile release of antigenic material at different time scales.

The research of LbL vaccine particles with a time-controlled pulsatile release has also been developed in the context of the SARS-CoV-2/COVID-19 pandemic outbreak. Zhou et al. [102] proposed the fabrication of a single injection COVID-19 vaccine based on LbL particles. For this purpose, they proposed the assembly of LbL particles by conjugating the antigen, S1 subunit of SARS-CoV-2 spike protein, in CaCO3 microspheres, followed by coating with a multilayer obtained by the alternate deposition of tannic acid (TA) and polyethylene glycol (PEG). This allows obtaining single-injection vaccines based on the mixture of particles with different number of bilayers. Thus, considering that the multilayers constructed by TA and PEG undergo erosion under physiological conditions at a fixed rate, the combination of particles including different number of layers ensure a release of the antigenic molecules following a pulsed profile which can be considered similar to that expected in multiple-dose vaccines. The results of the immunization using this type of vaccines were very promising, leading to potent and persistent S1-specific humoral and cellular immune responses in mice. Moreover, this type of vaccine ensures an immune response and viral inhibition that is comparable to that found for multiple-dose vaccines due to the release profiles based in the existence of different pulses. Figure 3 represents a scheme of the release profile of multi-pulse vaccines. Multi-pulse vaccines based on LbL particles have also been tested for the release of ovalbumin following similar concepts to the above discussed [101].

Di et al. [103] extended the design of single-dose vaccines with the aim to obtain self-boosting vaccination. For this purpose, they fabricated core-shell structures based on a core of microporous calcium carbonate particles loaded with poly(L-glutamic acid) containing a specific antigen (ovalbumin) forming the so-called “antigen-core” which is coated by the alternate deposition of antigen layers and an oppositely charged polypeptide poly(L-lysine) (PLL) forming the “antigen shell”. These structures are characterized by a three-phase release, inducing an effective recruitment of APCs and the efficient uptake of antigens. Moreover, this type of vaccine induces a strong secretion of chemokines and cytokines, which is very important for controlling the proliferation of splenocytes and the activation of T cells. On the other hand, the antigen release triggers the production of primary antibodies, which is strongly stimulated as the core-shell particles are degraded leading to the antigen release. It should be noted that this type of vaccine ensures an elicitation of antibodies that is faster than that obtained using traditional aluminum adjuvant vaccines. Therefore, this type of vaccines presents as a main advantage their ability for ensuring a quick production of antigen-specific antibodies, which reduces the probabilities of infection. Figure 4 shows a schematic representation of the fabrication process, and performance of a self-boosting core-shell microparticle vaccine.

A very promising alternative to vaccination is the intracellular delivery of nucleic acids. For this purpose, Czuba-Wojnilowicz et al. [104] proposed the design of LbL particles, both core-shell and hollow one, formed by the assembly of poly(arginine) and poly(4-styrene sulfonate of sodium), and loaded with a siRNA (small interfering RNA), which makes it possible the effective siRNA deposition and cell uptake. In fact, the fabricated particles deliver transcriptional gene silencing-inducing siRNA into the nucleus of HeLa T4^+^ cells infected with HIV, CD4^+^ T cells, resting primary CD4^+^ T cells, and monocytederived macrophages (MDM), contributing to the HIV-A gene silencing. Vaccination with genetic material was also explored by Xu et al. [105]. For this purpose, they built multilayers capsuled by the alternate deposition of mannosylated chitosan and a charge-reversible polymer poly (allylamine hydrochloride)-citraconic anhydride on a core formed by the complexation of a DNA plasmid with an amphiphilic dendritic lipopeptide (DSPE-G2). The latter contributes to DNA plasmid condensation, whereas the use of a charge reversible polymer in the shell facilitates the release of the DNA. The use of this type of non-viral vector demonstrated a good transfection efficiency, making LbL layers loaded with nucleic acid a very promising tool for vaccine fabrication.

Afford et al. [106] proposed the design of DNA vaccines by protecting the nucleic acid (7 kDa G-quadruplex DNA) in a stimuli responsive LbL hydrogel capsules formed by combining poly(methacrylic acid) and poly(*N*-vinylpyrrolidone). These capsules can release their cargo on demand under the action of specific enzymes or as response to ultrasounds. Moreover, the designed capsules can also encapsulate and release ~450 kDa double stranded DNA.

The use of capsules containing antioxidants is also a promising approach for eliciting specific immune response against specific diseases. Kozlovskaya et al. [107] fabricated LbL capsules assembled through hydrogen bonds between tannic acid (natural polyphenol) and two different polymers, poly(*N*-vinylpyrrolidone) and poly(*N*-vinylcaprolactam). The administration of these capsules contributes to the dissipation of proinflammatory reactive oxygen and nitrogen species, attenuating the production of interferon (IFN)-*γ* and tumor necrosis factor (TNF)-*α* proinflammatory cytokines due to the activity of cognate antigen-stimulated autoreactive CD4^+^ T cells. This suggests that capsules containing tannic acid may contribute to a very efficient immunomodulatory response. The use of antioxidant combined with an antigen such as ovalbumin can contribute to stimulate specific immune response as was probed by Feduska et al. [108]. They encapsulated ovalbumin on multilayers of poly(*N*-vinylpyrrolidone) and tannic acid (TA) which contribute to elicit a strong immune response in LPS-activated dendritic cells. Moreover, upon administration in mice, the capsules elicited a decrease in CD4 T cell differentiation and effector responses. Therefore, this type of capsule can effectively blunt innate immune-derived proinflammatory third signal synthesis.

The above discussion suggests that the use of LbL particles, in absence of any adjuvant or formulation additive, may be a very promising strategy to produce vaccines against a broad range of diseases. In fact, microparticle vaccines may help to increase the vaccination rates, preventing immunization losses, and minimizing economic and time costs.

## 5. Vaccines Based on Noble Metal Particles Decorated with LbL Films

Noble metal particles have been extensively exploited for different applications in nanomedicine taking advantage of their immunological inert character and reduced toxicity. Moreover, this type of particle can be synthetized with well-controlled properties, including their size or shape [109,110]. This makes their use as vaccination platforms possible upon their modification with antigenic compounds [111,112]. It may be expected that the use combination of supramolecular systems based on noble particles decorated with LbL films may contribute to the design of a new vaccine type, enabling a rational control over the immune responses.

Zhang et al. [113] proposed the fabrication of immune-polyelectrolyte multilayers (iPEMs) by the alternate deposition on gold nanoparticles of a peptide antigen (cationic version of the SIINFEKL peptide, SIIN*) and the polyinosinic-polycytidylic acid (polyIC), a polyanionic toll-like receptor (TLR) agonist, which acts as adjuvant. This type of supramolecular systems can be easily internalized by dendritic cells (DCs), leading to an adjuvant-controlled activation of CD40^+^, CD80^+^, and CD86^+^ surface markers, selective triggering of TLR signaling, and release of the antigens contained within the iPEMs. Moreover, iPEMs induce an important proliferation of antigen-specific T cells and effector cytokines, avoiding the production of cytokines resulting from inflammatory reactions. In fact, the immunization with iPEMs promotes a high concentration of antigen specific CD8^+^ T cells in peripheral blood after 1 week from the administration. Figure 5 presents a sketch of the preparation process of iPEMs and the mechanism of immune response production. iPEMs should be considered an excellent alternative to transport different immune cargos for vaccination purposes.

Zhang et al. [114] extended their study of iPEMs based in LbL films on noble metal nanoparticles to the study of the role of the dose, the administration route, and the type of molecular adjuvant in the induction of T cell immunity. For this purpose, three different routes of administration (intradermal, subcutaneous, and intramuscular) and three different doses were tested. The results demonstrated that the use of intradermal administration induces a stronger immune response, there being a dependence of the intensity of the immune response on the administered dose. Moreover, it was reported that this type of vaccination platforms can induce a durable immune memory as evidenced by the potent antigen specific CD8^+^ T recall responses after 49 days of the vaccine administration. On the other hand, the type of adjuvant was also found to be very important in the level of immune response. In fact, iPEMs containing CpG (TLR9 agonist) as adjuvant induce a stronger immune response than those vaccines containing polyIC.

Bishop et al. [15] proposed the design of LbL-decorated gold nanoparticles loaded simultaneously with DNA and siRNA. These platforms were easily internalized by human primary brain cells where reaching the cytoplasm and the nucleus, they deliver simultaneously the DNA and the siRNA. Thus, it is possible to induce two effects simultaneously: (i) gene expression, and siRNA-mediated knockdown. The latter occurs with higher efficiency than that observed for commercially available transfection reagents.

The use of supramolecular systems based on noble metal nanoparticles decorated with LbL layers as vaccines may be exploited for the delivery of a broad range of immune cargos to stimulate the immune response against different diseases. However, their true translational application remains challenging due to different drawbacks to the use of noble metal nanoparticles in vivo, including their size and dose dependent toxicity [11].

## 6. LbL Microneedles for Vaccination

The fabrication of microneedles (MNs) based on LbL nanoassemblies presents a big interest for the fabrication of vaccination platforms densely loaded with different biological cargoes [115]. However, the long epidermal application time required for drug implantation when MNs are used has been an important drawback for the development of this type of platform on vaccine administration, even though they provide a strategy enabling a pain-free self-administration. Moreover, MNs reduce the potential risks associated with the reutilization of needles, accidental injuries provoked by an erroneous manipulation of conventional needles, and bloodborne pathogen transmission [116,117,118]. Therefore, the optimization of fabrication of non-dissolvable microneedles enabling a fast and complete lift-off strategy is necessary. This challenge can be, at least partially, faced by using LbL-based nanomaterials, as was demonstrated by He et al. [116]. They fabricated LbL microneedle systems based on the assembly of a synthetic charge-invertible polymer, poly(2-(diisopropylamino) ethyl methacrylate-b-methacrylic acid) (PDM). This polymer undergoes a pH-modulated charge inversion, forming cationic micellar aggregates under acidic conditions which switch to negatively charged aggregates upon the increase of the pH up to values close to physiological ones. The assembly of a PDM layer, adsorbed under conditions where it remains positively charged (pH = 5.4), directly attached to the substrate (a negatively charge poly(L-lactic acid) (PLLA) patch containing 77 pyramidal microneedles) followed by the construction of the LbL film by the alternate deposition of layers of poly(*N*″{*N′*-[*N*-(3-aminopropyl)-2-aminoethyl]-3-aminopropyl}aspartamide) (PAsp(EDDPA)) and the model antigen chicken ovalbumin allows manufacturing LbL MNs which enable a fast implantation. In fact, this type of device requires only 1 min insertion into the skin to ensure the implantation (see Figure 6 where the amount of delivered ovalbumin at different times after implantation can be observed as result of their labeling with fluorescent moieties). The action mechanism of this type of MN is based on the release of the LbL film as result of the repulsion occurring between the PDM layer, which adopts a negative charge once the MNs enter in contact with the physiological medium during the implantation process, and the negatively charged adjacent layer and substrate. Figure 6 displays a schematic representation of the architecture of the assembled systems and the implantation process. The above structures ensure a sustained release of the antigen for 3 days, which contributes to an elevated immune response characterized by a higher humoral immunity than the obtained via conventional administration routes. In fact, the level of serum ovalbumin specific IgG_1_ (immunoglobulin G_1_) for the group of mice vaccinated using a microneedle-based strategy was 9-fold and 160-fold higher than that found when the vaccination occurs using intramuscular and subcutaneous injection methods, respectively. Moreover, the use of the manufactured MNs gives as result an efficient immune activation characterized by the fast vaccine adjuvant uptake by the antigen existing in the cells.

De Muth et al. [92] fabricated a non-invasive vaccine formulation based on the coating of poly(lactide-co-glycolide) microneedle arrays (decorated with 20 bilayers of protamine sulfate (PS) and sulfonated poly(styrene) (SPS)) with a LbL structure formed by a biodegradable cationic poly(β-amino ester) (Poly-1) and negatively charged interbilayer-cross-linked multilamellar lipid vesicles (ICMVs) loaded with a protein antigen and the molecular adjuvant monophosphoryl lipid A. The in vivo application of this type of platforms to the skin of mice for 5 min confirmed the fast transference of this type of vaccination platforms from the surface of the NMs into the cutaneous tissue. Moreover, the implanted formulation remained in the skin after the removal of the MNs. Thus, a sustained release of the cargoes over 24 h was possible, which makes the uptake of the lipid nanocarriers by the antigens existing in the cells (APCs) possible, triggering their activation. This type of vaccination strategy promotes a robust antigen-specific humoral immune response, resulting in a balanced generation of immunoglobulins. The main advantage of the LbL multilayers containing vesicles within the delivery platform Is related to their immunogenicity when they are delivered to the skin. Figure 7 shows a scheme including the structure of the obtained vaccination platforms with adsorbed vesicles within the LbL structure, and their action mechanism upon injection.

Uppu et al. [119] proposed the use of LbL microneedles for the administration of a three component subunit vaccine including a dengue virus Envelope protein Domain III (EDIII) subunit antigen and two adjuvants, a double-stranded RNA (Poly (inosinic:cytidylic acid) (PolyI:C)), and an amphiphilic hexapeptide, Pam3CSK4. The administration of this type of vaccine allows a sustained release of the different vaccine components. This occurs through specific release profiles, which can be tuned by changing the composition and structure of the LbL multilayer. Thus, it is even possible to independently control the release of each single component in time scales ranging from days up to two weeks.

van der Maaden et al. [120] designed microneedle vaccines for the virus of polio. For this purpose, they fabricated a LbL coating on microneedle arrays (initially decorated with pH-sensitive pyridine groups) by the deposition of 10 bilayers formed by alternate layers of inactivated polio virus (IPV) particles and *N*-trimethyl chitosan chloride (TMC). This makes it possible to ensure an antigen dose high enough for vaccination purposes. The in vivo application of the fabricated microneedle arrays in rats was accompanied by the induction of IPV-specific antibody responses. For polio virus, very different immune responses were found depending on the used antigenic form. The use of microneedles loaded with the C-antigen (expressed on noninfectious virus particles) led to a high and robust antibody production which is equivalent to the immunization obtained upon intramuscular vaccination. However, the use of microneedles including the D-antigen (expressed in native polio virus) produced smaller immune response than that obtained after intramuscular vaccination, which suggests a possible virus denaturation during the coating process. Therefore, it is possible that the effectiveness of microneedles against the polio virus can be only ensured by stabilizing the antigenic material before its inclusion in the supramolecular nanostructure obtained by the LbL method. A similar strategy to the used for design microneedles against polio virus was exploited by Schipper et al. [121] for fabricating vaccines against diphtheria. They combined layers of TMC and diphtheria toxoid to coat pH-sensitive microneedles and tested the ability of microneedles coated with different number of bilayers (2, 5, and 10) to produce an immune response. The results showed that the increase of the bilayer number induced a stepwise increase of the specific immune response. In particular, the immunization with microneedles coated by 10 bilayers produced an immunity similar to that obtained upon subcutaneous injection with a diphtheria toxoid dose of about 5 μg. This is equivalent to an 8-fold decrease of the dose required for immunization when LbL materials are used (diphtheria toxoid dose about 0.6 μg in LbL based microneedles against the 5 μg required in subcutaneous immunization for the same immune response).

Kim et al. [122] designed microneedles for DNA vaccination formed by the alternate deposition of heparin and albumin layers on the surface of microneedles coated with polydopamine. These structures were coated with a layer formed by polyplexes formed by the association of poly(ethylenimine) (PEI) of low molecular weight (1.8 kDa), mannose (Man), deoxycholic acid (DA), and a DNA plasmid, and a layer N-formyl-methionyl-leucyl-phenylalanine microspheres (fMLP-MS) which can act as a chemoattractant for APCs which contribute to the enhancement of the DNA, improving patient compliance. The response of the multilayers to the change of pH upon their injection allows the uptake of the polyplexes by the dermal epithelial cells and resident APCs where the expression and secretion of the antigen occurs. In fact, the introduction of polyplexes contributes to an enhanced delivery of the DNA vaccine at the cellular level, encoding a secretable Aβ_1→42_ as an antigen. Moreover, the sustained release of the fMLP-MS helps the recruitment of immune cells, e.g., macrophages, close to the injection site. This leads to an efficient uptake and process of the secreted antigen to present epitopes of major histocompatibility molecules. This contributes to increasing the immune response relative to conventional hypodermic vaccines. Figure 8 shows a scheme including the structure of the microneedles used for vaccination, and the vaccination process.

De Muth et al. [17] also exploited the ability of DNA plasmids as suitable material for the stimulation of the immune response. They prepared microneedles coated with 35 bilayers obtained by the alternate deposition of a biodegradable poly(*β*-amino-ester) and different DNA plasmids. These types of vaccines emerge as good alternatives to maintain the bioactivity of the genetic material, producing upon administration a strong immune response against a model HIV antigen and enhanced memory for the generation of T-cells. Moreover, this vaccination approach elicits a gene expression that is about 140-fold higher than that obtained upon intradermal injection of DNA vaccines, which opens promising avenues for the use of microneedles for DNA vaccination.

In summary, the use of LbL microneedles is a powerful alternative to enhance the delivery of vaccines, providing an effective platform to guarantee a proper distribution of the vaccination agents and their stability during storage.

## 7. Conclusions

Vaccination is currently the most powerful strategy to minimize the risk associated with the emergence of pandemic outbreaks of different diseases, reducing their mortality. However, many times it is very difficult to find suitable platforms ensuring their therapeutic efficiency and patient convenience, as well as their worldwide accessibility. The latter has become a very critical issue for reducing the impact of some diseases, especially in low-income countries where the accessibility to vaccination programs is very limited. In this context, the versatility and modularity of the Layer-by-Layer (LbL) method open new avenues for designing vaccination platforms with high efficiency and specific targeting, minimizing the economic impact associated with their fabrication. Moreover, the flexibility of LbL nanoassemblies offers important advantages to prepare a well-sketched platform with tailored properties to guarantee durable and functional immune responses, which opens very interesting perspectives for vaccination against clinically relevant diseases.

Despite the promising framework offered for translational applications, it is mandatory to deepen the understanding of the release mechanism of LbL vaccines, and how to control it for clinically relevant applications. Moreover, the transition from laboratory to clinical practice requires a significant amount of work on screening by using human relevant models. Facing these issues will make it possible to obtain relevant bioengineered LbL materials, which can be exploited as platforms for clinical vaccination platforms. The avenue is open to the research and development of LbL materials as a tool for obtaining most effective vaccines under clinically relevant conditions. However, most of the results obtained to date have been limited to animal models, which, even if satisfactory, do not allow the safety and efficacy of LbL vaccines to be determined in humans. In addition, the translation of LbL vaccines from animal models to clinical use is complicated by several factors. LbL vaccines need to be optimized in terms of formulation, dosage, and delivery methods. This must consider that the human immune system is diverse, and the optimal formulation and delivery method may vary depending on the specific population or disease being targeted. This variability makes it challenging to develop a universal LbL vaccine. On the other hand, the regulatory process for vaccines is rigorous, and LbL vaccines must demonstrate both safety and efficacy in clinical trials before they can be approved for use. This process can be time consuming and resource intensive, requiring significant investment in research and development. Finally, the shelf life and storage requirements of LbL vaccines can pose logistical challenges, particularly in resource-limited settings.

## Figures and Tables

**Figure 2 pharmaceutics-15-01449-f002:**
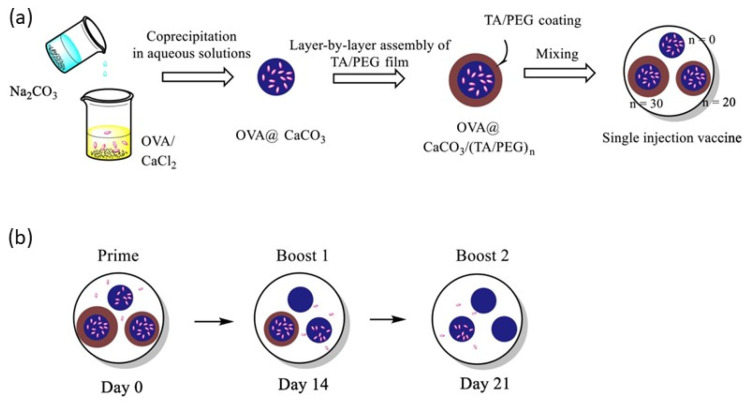
(**a**) Sketch representing the fabrication of single-injection vaccines based on LbL structures. (**b**) Sketch of the time-controlled pulsatile release of antigenic material at different timescales. Adapted from Wang et al. [101]. Copyright (2022), with permission from Elsevier.

**Figure 3 pharmaceutics-15-01449-f003:**
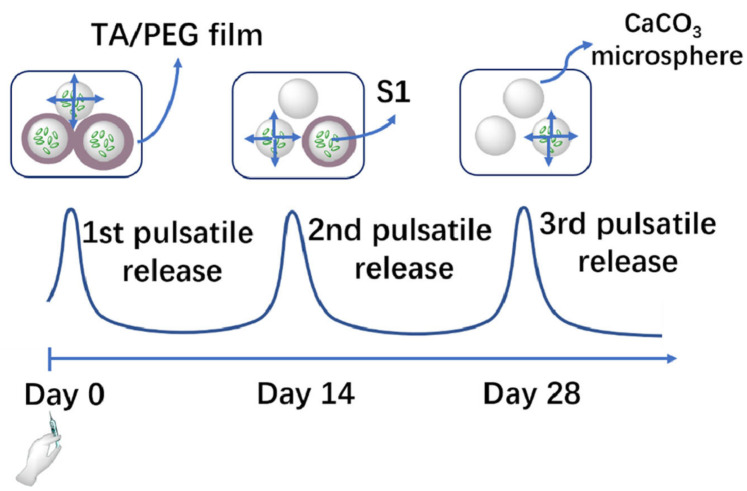
Sketch representing the release profile including several pulses of vaccine against SARS-CoV-2 using LbL particles. Reprinted from Zhou et al. [102]. Copyright (2022), with permission from Elsevier.

**Figure 4 pharmaceutics-15-01449-f004:**
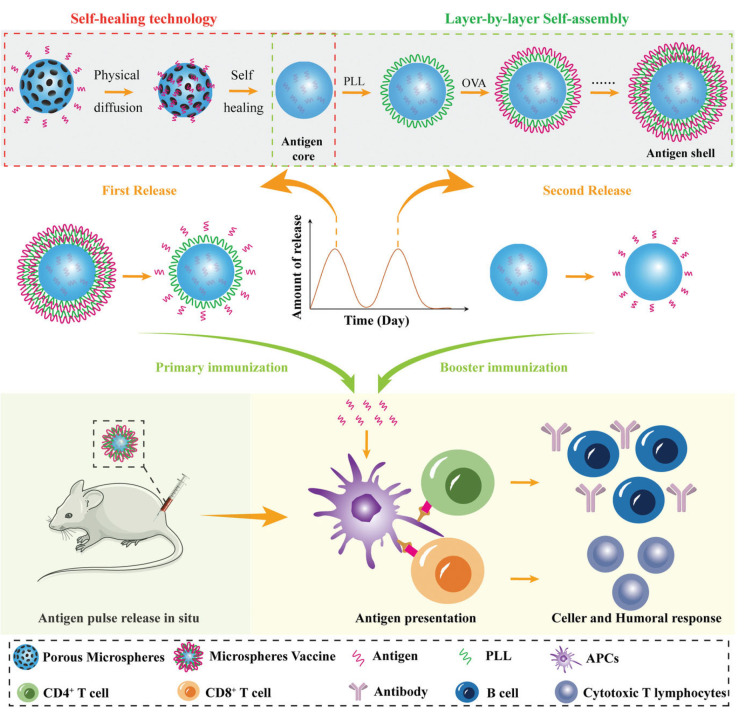
Sketch representing the fabrication process and action mechanism of self-boosting core-shell microparticle vaccines. Reprinted from Di et al. [102,103]. Copyright (2023), with permission from John Wiley and Sons.

**Figure 5 pharmaceutics-15-01449-f005:**
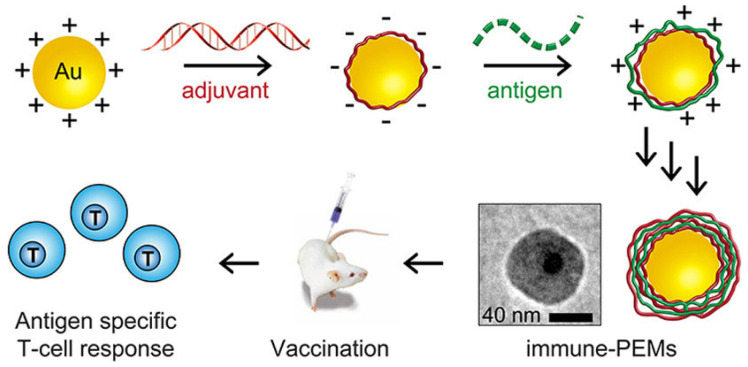
Sketch representing the fabrication process of iPEMs and their action mechanism of self-boosting core-shell microparticle vaccine. Reprinted from Zhang et al. [113] under CC-BY license.

**Figure 6 pharmaceutics-15-01449-f006:**
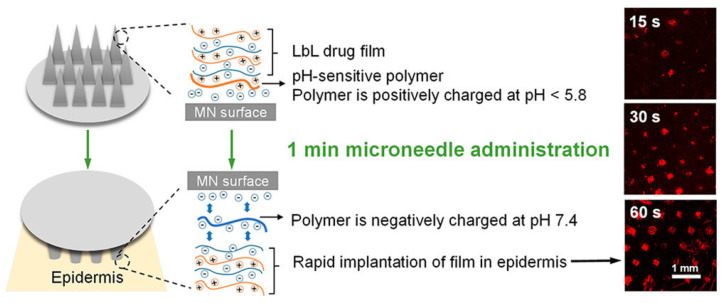
Sketch representing the structure of the assembled systems and the implantation mechanism together with confocal microscopy images where the delivery of fluorescent-labeled ovalbumin to mouse ear skin after 15 s, 30 s, and 60 s of microneedle insertion, respectively, is evidenced. The scale bars correspond to 1 mm. Reprinted with permission from He et al. [116]. Copyright (2018) American Chemical Society.

**Figure 7 pharmaceutics-15-01449-f007:**
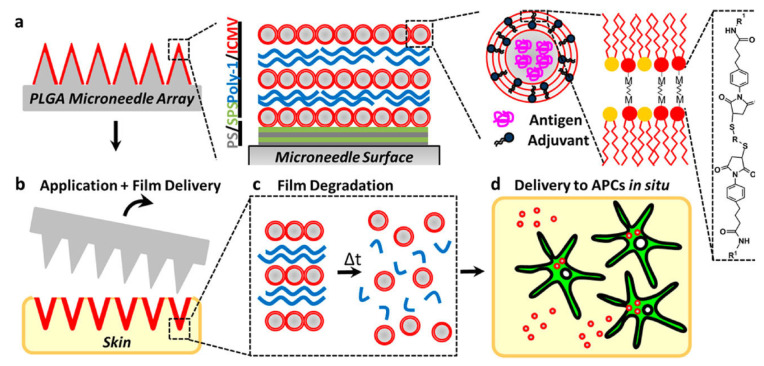
(**a**) Sketch of the (Poly-1/ICMV) multilayers decorated microneedle surfaces. (**b**) Microneedle application into the skin. (**c**) Hydrolytic degradation of Poly-1 releasing the ICMV. (**d**) ICMV delivery to skin APCs. Reprinted with permission from DeMuth et al. [92]. Copyright (2012) American Chemical Society.

**Figure 8 pharmaceutics-15-01449-f008:**
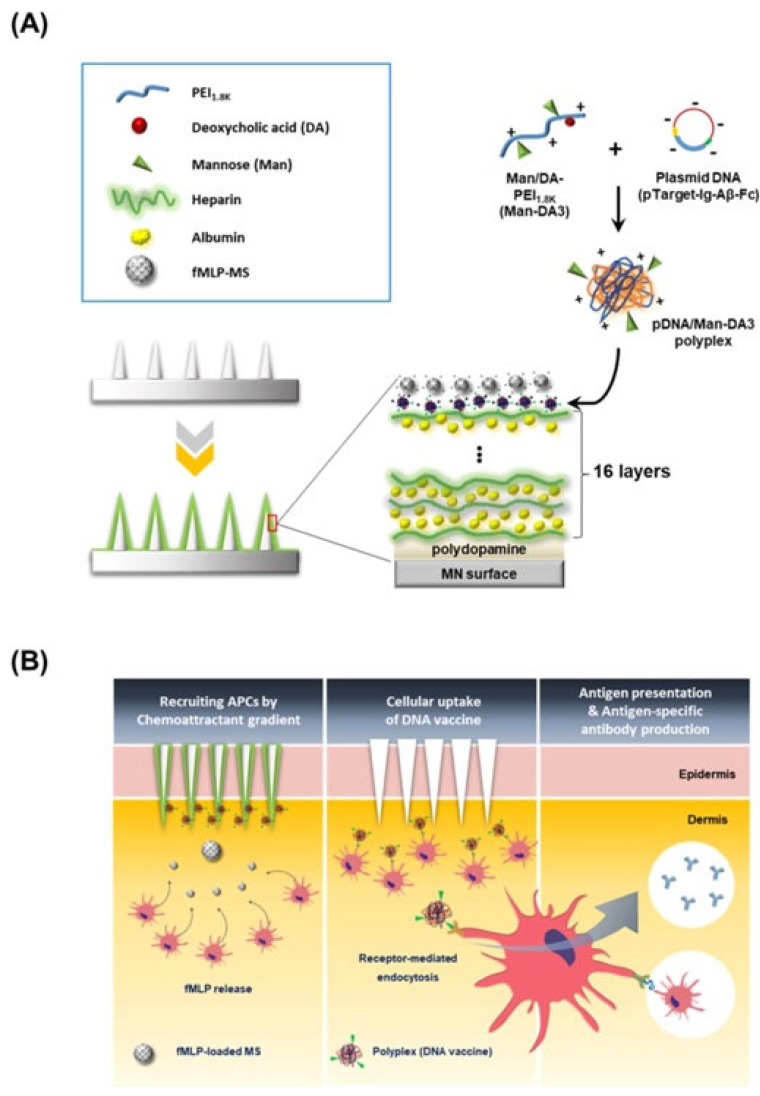
(**A**) Sketch representing the structure of LbL microneedles loaded with polyplexes and chemoattractant. (**B**) Sketch of the vaccination process. Reprinted from Kim et al. [122]. Copyright (2023), with permission from Elsevier.

**Table 1 pharmaceutics-15-01449-t001:** Summary of the most common strategies used for the fabrication of LbL materials depending on the nature of the substrate.

Type of Substrate	Fabrication Method	References
Flat	Dipping	[19]
	Spin coating	[39]
	Spray-assisted deposition	[40,41]
	High gravity field assisted deposition	[42]
	Simultaneous spray coating of interacting species (SSCIS)	[43,44]
	Electric/Magnetic field assisted deposition	[45,46]
Colloidal	Immersive deposition (depends on the substrate nature)	[31,33,34,47,48,49,50,51,52,53,54]
	Magnetic field assisted deposition (only for magnetic colloid)	[55]
	Deposition on immobilized colloid	[56]
	Continuous deposition using tubular flow reactors	[57]
	Microfluidic assisted fabrication	[58,59,60,61]
	Fluidized bed deposition	[62]

## Data Availability

This work did not result in the production of new data.

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
