# Peer review of "Layer-by-Layer Nanoassemblies for Vaccination Purposes"

_pharmaceutics, 2023, doi:10.3390/pharmaceutics15051449_

Round 1

Reviewer 1 Report

this looks like a solid review on the current state of the field for LbL systems in the application area of vaccines. the authors are established in this field and know the technology well, having made contributions to the advancement of LbL systems themselves. 

the only recommendation is the consideration of a few more references from the Kharlampieva and Tse groups who have both worked with and studied antioxidant LbL systems for immunomodulation. 

Author Response

Reviewer 1

This looks like a solid review on the current state of the field for LbL systems in the application area of vaccines. the authors are established in this field and know the technology well, having made contributions to the advancement of LbL systems themselves. 

the only recommendation is the consideration of a few more references from the Kharlampieva and Tse groups who have both worked with and studied antioxidant LbL systems for immunomodulation. 

We have added some additional references related to the topic cited by the reviewer.

We thank to the reviewer for his/her comments, they were very useful for improving our manuscript.

Reviewer 2 Report

Guzman et al. wrote a very interesting and comprehensive review of the potential use of LbL for vaccination.

The conclusion section would benefit for a more extensive discussion on the current gaps preventing going from proof-of-principle in animal models (as extensively discussed in the different sections of the present review) to clinical evaluation and eventually implementation.

Additional comments:

Section 3: While I agree that skin patch for vaccine delivery has not been so promising, the sublingual approach as described by Monge et al. should not be discarded. Please modify the introduction paragraph of section 3 to not only categorize flat film LBL approaches as proof of principles.

Section 4: 

Line 214-215: I assume the authors used a recombinant Listeria expressing ovoalbumine. Please specify.

Line 216-218: I don’t understand the point the authors want to make here. Please clarify.

Section 6:

Line 501-502: sentence seems incomplete

Author Response

Guzman et al. wrote a very interesting and comprehensive review of the potential use of LbL for vaccination.

The conclusion section would benefit for a more extensive discussion on the current gaps preventing going from proof-of-principle in animal models (as extensively discussed in the different sections of the present review) to clinical evaluation and eventually implementation.

Following the reviewer recommendation, we have extended the discussion in the conclusion section.

Additional comments:

Section 3: While I agree that skin patch for vaccine delivery has not been so promising, the sublingual approach as described by Monge et al. should not be discarded. Please modify the introduction paragraph of section 3 to not only categorize flat film LBL approaches as proof of principles.

We thank to the reviewer for the comment, we have modified the text according.

Section 4: 

Line 214-215: I assume the authors used a recombinant Listeria expressing ovoalbumine. Please specify.

Unfortunately, this information is missing in the original reference.

Line 216-218: I don’t understand the point the authors want to make here. Please clarify.

We have revised-

Section 6:

Line 501-502: sentence seems incomplete

Actually, the sentence does not provide any information. We have removed it.

We thank to the reviewer for his/her comments, they were very useful for improving our manuscript.

Reviewer 3 Report

Dear authors,

Please respond to:

1) Line 43: Perhaps substitute the word 'immune' with 'immunogenic' so that it means that the cargo is immune response-eliciting content.

2) Under heading 2. A brief introduction to the LbL method. Could you add some information regarding the interpenetration of different layers?

3) Line 126: Correct 'column' to 'columns'.

4) Line 164: Correct 'ovoalbumin' to 'ovalbumin' throughout the whole text.

5) Line 183: Do you think that the immunogenic material should always be encapsulated? Can they simply be coated onto empty LbL carriers? One could, for example, inject the material intramuscularly to avoid gastric and hepatic decomposition etc.

6) You mention several proteins, peptides, poly(amino acid)s as coating materials. Did you find any other biopolymers that could be used?

7) Line 364: Noble metals are very expensive and rare. Could you find any examples of non-noble metals being used instead?

8) Line 424: Check the spelling of 'blood-borne'.

9) According to IUPAC rules, heteroatoms should be written in italics. For example 'N-', 'O-' and 'S-' etc.

Author Response

Reviewer 3

Dear authors,

Please respond to:

1) Line 43: Perhaps substitute the word 'immune' with 'immunogenic' so that it means that the cargo is immune response-eliciting content.

We agree with the reviewer, and we change the term for the correct form.

2) Under heading 2. A brief introduction to the LbL method. Could you add some information regarding the interpenetration of different layers?

A brief statement about the interpenetration between the different layers is now included in the revised version.

3) Line 126: Correct 'column' to 'columns'.

We have corrected in the text.

4) Line 164: Correct 'ovoalbumin' to 'ovalbumin' throughout the whole text.

We have corrected in the whole text.

5) Line 183: Do you think that the immunogenic material should always be encapsulated? Can they simply be coated onto empty LbL carriers? One could, for example, inject the material intramuscularly to avoid gastric and hepatic decomposition etc.

The comment of the reviewer is correct, we have included a brief statement about that issue.

6) You mention several proteins, peptides, poly(amino acid)s as coating materials. Did you find any other biopolymers that could be used?

On the best of our knowledge, we have included in the text the most common material used for the fabrication of LbL layers for vaccination purposes.

7) Line 364: Noble metals are very expensive and rare. Could you find any examples of non-noble metals being used instead?

On the best of our knowledge, LbL materials for vaccination purposes have been fabricated only by including noble metals, we do not have any knowledge about the use of any other type of metal particles.

8) Line 424: Check the spelling of 'blood-borne'.

We have revised the text.

9) According to IUPAC rules, heteroatoms should be written in italics. For example 'N-', 'O-' and 'S-' etc.

We have revised for correctness.

We thank to the reviewer for his/her comments, they were very useful for improving our manuscript.

Reviewer 4 Report

The manuscript ‘Layer-by-layer nanoassemblies for vaccination purposes’ presents a very interesting overview of the use of LbL technology for vaccine design. The review is well written, organized and illustrated. I only have few minor comments:

- There are some spell check required (line 63, 82, 425…)

- p6, line 248: the type of immunoglobulin is missing (G). It is correctly written p450.

- The name of the model antigen Ovalbumin is wrong all over the manuscript (ovoalbumin).

- What is the difference between the scheme in fig 2b and the figure 3?

- line 338: ‘A very promising alternative for vaccination purposes…’ is presented through an siRNA delivery system for treatment against HIV. However, gene silencing is not considered as vaccination. Did you want to say ‘an alternative to vaccination’?

- p14: replace figure 6 by figure 8. Does the mannose correspond to grey or green arrowheads?

Author Response

Reviewer 4

The manuscript ‘Layer-by-layer nanoassemblies for vaccination purposes’ presents a very interesting overview of the use of LbL technology for vaccine design. The review is well written, organized and illustrated. I only have few minor comments:

- There are some spell check required (line 63, 82, 425…)

Revised

- p6, line 248: the type of immunoglobulin is missing (G). It is correctly written p450.

We have revised the text.

- The name of the model antigen Ovalbumin is wrong all over the manuscript (ovoalbumin).

We have corrected in the whole text.

- What is the difference between the scheme in fig 2b and the figure 3?

We agree with the reviewer, the inclusion of both may appear as a repetition. However, Figure 3 states better the concept of the pulsed release. We prefer to maintain both.

- line 338: ‘A very promising alternative for vaccination purposes…’ is presented through an siRNA delivery system for treatment against HIV. However, gene silencing is not considered as vaccination. Did you want to say ‘an alternative to vaccination’?

We agree with the reviewer, and we have corrected in the text.

- p14: replace figure 6 by figure 8. Does the mannose correspond to grey or green arrowheads?

We have corrected. Mannose is now homogeneously represented in the text.

We thank to the reviewer for his/her comments, they were very useful for improving our manuscript.